# Functions of mHealth Diabetes Apps That Enable the Provision of Pharmaceutical Care: Criteria Development and Evaluation of Popular Apps

**DOI:** 10.3390/ijerph20010064

**Published:** 2022-12-21

**Authors:** Bushra Ali Sherazi, Stephanie Laeer, Svea Krutisch, Armin Dabidian, Sabina Schlottau, Emina Obarcanin

**Affiliations:** 1Institute of Clinical Pharmacy and Pharmacotherapy, Heinrich Heine University, Universitätsstraße 1, 40225 Düsseldorf, Germany; 2Institute of Pharmacy, Faculty of Pharmaceutical and Allied Health Sciences, Lahore College for Women University, Lahore 54000, Pakistan; 3Meala GmbH, Dolziger Str. 7, 10247 Berlin, Germany; 4Department of Pharmacy, National University Singapore, 18 Science Drive 4, Singapore 117559, Singapore

**Keywords:** pharmaceutical care, mobile apps, diabetes apps, mHealth, adherence, patient preferences, medication management, interoperability

## Abstract

Personal digital health apps for managing diabetes should include functions that enable the provision of pharmaceutical care services and allow within-app communication with pharmacists and other healthcare providers, thereby improving patient outcomes. The primary aim of this study was to assess the functions of diabetes apps that were relevant to providing pharmaceutical care services (i.e., medication management, adherence, non-pharmacological management, interoperability, and communication). Sixteen criteria related to pharmaceutical care were developed and then used to assess ten popular diabetes apps. The highest numbers of pharmaceutical care criteria were met by the apps Diabetes:M and mySugr (11 criteria); Contour™Diabetes, Dario Health, and OneTouch Reveal^®^ (ten); and DiabetesConnect and ESYSTA (nine); followed by Glucose Buddy (eight), meala (seven), and lumind (three). The most prevalent functions were related to promoting adherence and non-pharmacological management, but most criteria relevant to medication management were lacking. Five apps allowed within-app communication between patients and healthcare professionals (HCPs); however, no app included communication with pharmacists. High-quality diabetes apps are powerful tools to support pharmaceutical care and remotely monitor diabetes patients. Improvements are needed as they often lack many medication management functions, including within-app communication with HCPs (especially pharmacists). To maximize diabetes app use and improve outcomes, app developers should consider including pharmacists alongside other healthcare providers when customizing app designs.

## 1. Introduction

Digital health solutions, particularly mobile health applications (mHealth apps), are increasingly being used to monitor patients remotely and deliver healthcare services outside conventional healthcare settings. The recent COVID-19 pandemic further intensified the need for digital health services [1].

It is necessary to choose carefully among the widely available mHealth apps, including disease-specific apps, as concerns exist regarding their reliability, data privacy and security, suitability of use, and clinical benefits [1,2,3]. Regulatory assessments and approvals are still not available for the majority of available mHealth apps in most countries [4]. In 2019, Germany became the first country worldwide to introduce statutorily reimbursable “apps on prescription” (Digitale Gesundheit Anwendungen, DiGA) under the Digital Healthcare Act (Digitale Versorgung Gestzt—DVG) and the Digital Health Applications Ordinance (Digitale Gesundheitsanwendungen Verordnung—DiGAV) [5]. mHealth apps undergo a strict evaluation process to qualify for inclusion in the official DiGA directory [5]. The General Federal Institute for Drugs and Medical Devices (Bundesamt für Arzneimittel und Medizinprodukte—BfArM) approved the first DiGA in October 2020, and as of 21 November 2022, it has approved a total of 33 DiGAs for different indications [6,7]. Since the approval of the first DiGA, 50,100 DiGAs have been prescribed or approved directly by health insurers in Germany [8].

The integration of mHealth apps into clinical practice, however, requires much greater support by healthcare professionals (HCPs) in order to optimize the effectiveness of these apps [9,10]. Pharmacists, as the most accessible HCPs, are in an ideal position to promote awareness and the effective use of digitally assisted health support in the form of mHealth apps [11,12]. In addition, interactive mHealth interventions have the potential to improve pharmaceutical care outcomes by supporting contact between pharmacists and patients [13,14]. mHealth apps in the DiGA directory can only be prescribed by physicians and psychotherapists or approved directly by health insurers at the patient’s request; however, two surveys of German physicians have reported very low rates of prescribing of mHealth apps [15,16]. Adherence to mHealth apps has also been reported to be suboptimal, with only 78% of approved or prescribed DiGAs having been activated by patients [8]. Hence, pharmacists could play an important role in recommending mHealth apps to patients and increasing adherence to their use. Data from England suggested that 56% of respondent pharmacists were aware of mHealth apps, and 60% of those recommended apps to patients [12].

Recently, a boom has been occurring in the global market of digital apps for diabetes patients [17], with diabetes apps accounting for 15% of the total number of disease-specific apps in 2021 [1]. This high interest has been reflected in the increased usage of these apps by patients with diabetes [18]. An increasing body of evidence corroborates the effectiveness of mHealth apps in diabetes management [10,19,20]. A systematic review and meta-analysis of 13 interventional studies reported an improvement in glycated hemoglobin (HbA1C) values and diabetes self-management through the use of diabetes apps [21].

When pharmacists and other HCPs are considering which diabetes app to recommend, it is important that they consider the various capabilities of the apps. Studies have used a number of different terms to describe the wide range of app capabilities, such as app functionalities/functions, app characteristics, and app features [22]. A study by Smahel et al. used the term ‘functions’ as an overreaching term to describe the various app features which enable their users to select among a range of capabilities such as monitoring, setting goals, planning, providing feedback on performance, and communicating with other users [23]. Accordingly, ‘function’ is used in this evaluation to describe such features. To our knowledge, subcategories of app functions, especially those that are clinically relevant (i.e., pharmaceutical care criteria) have not been defined in the literature.

Selecting the appropriate mHealth diabetes app with adequate clinical functions for pharmaceutical care and to satisfy the technological needs of both HCPs and patients is critical. Currently, there is large variability in the key functions of diabetes apps, making it difficult to select the app that is most appropriate for an individual [24]. Salari et al. identified a minimum set of functions for diabetes mobile apps, which include the tracking of blood glucose, insulin and medication, physical activity, weight and body mass index, blood pressure, and diet; the provision of food databases, educational materials, and features that promote healthy coping, risk reduction, and problem-solving; the ability to message, color code, customize themes, set alerts, reminders, and target ranges and view trend charts, logbooks, and numerical indicators; and the inclusion of preset and custom notes [25].

Current diabetes apps focus on blood glucose monitoring, self-management, motivation for medication adherence, and lifestyle modifications [26,27,28]. Many of these functions including unique functionality to store and display data, to indicate trends and patterns in blood glucose and HbA1C values, and to track medication, diet and physical activity, allowing pharmacists to monitor patient therapy and intervene remotely when necessary. However, diabetes apps may lack some important functions, such as within-app interactions with HCPs and/or pharmacists, that could enhance the provision of pharmaceutical care. The inclusion of pharmaceutical care functions in diabetes apps is important, as they could enhance the development of individualized pharmaceutical care plans, thereby improving patient outcomes, as has been shown with the implementation of other pharmacist interventions in diabetes care [29,30,31,32]. The general technological functions of diabetes apps are also important, as are the functions that pertain to the preferences of the individual patient.

Although the functions of diabetes apps have been previously reviewed [26,33,34,35], to our knowledge, no study has evaluated the functions of currently available diabetes apps from a pharmaceutical care perspective. Therefore, in the present study we aimed primarily to provide information about the functions of popular diabetes apps relevant to providing pharmaceutical care services to patients with diabetes. The first major question—‘What functions of diabetes apps support the provision of pharmaceutical care?’—was answered by developing evaluation criteria based on the literature.

These criteria were then used to evaluate the functions of selected diabetes apps, thereby answering the second question—‘Which currently available diabetes apps fulfil the criteria relevant to pharmaceutical care?’. By answering the first two questions, this study addresses whether diabetes apps can be used as tools in pharmaceutical care and, if this is the case, provides a resource which pharmacists and other HCPs can use to better select apps based on their pharmaceutical care functions.

To give a more complete overview of the functions of diabetes apps, the final question addressed in this study was ‘What additional app functions should pharmacists consider to satisfy the technological needs and preferences of diabetes patients?’. This was answered by developing relevant evaluation criteria, which were used to evaluate the selected diabetes apps.

## 2. Methods

### 2.1. Developing the App Evaluation Criteria

#### 2.1.1. Criteria Relevant to Pharmaceutical Care

To answer the three study questions, diabetes app evaluation criteria first needed to be developed. Thus, EO, SL, BAS, AD, and SS (pharmacy faculty members of Heinrich-Heine-University Düsseldorf; HHU) defined app evaluation criteria using the three major study objectives: app criteria relevant to pharmaceutical care; general aspects of diabetes apps; and special app functions and patient preferences.

To the best of our knowledge, diabetes app functions have not previously been defined and explored from a pharmaceutical care perspective. Therefore, the authors (EO, BAS, AD, SS) performed an extensive literature search using the PubMed database. The literature search was conducted for papers published until January 2022, using the search terms “Pharmacy”, “digital apps”, “Mobile apps”, “mHealth”, “Pharmaceutical care”, and “adherence”. We compiled a list of essential criteria based on the important pharmaceutical care interventions for diabetes management and included three major categories: Medication Management, Adherence/Non-pharmacological Management, and Interoperability/Communication [12,13,36,37,38,39,40,41]. After initially defining the app criteria, the authors (EO, SL, BAS, AD, SS) tested the criteria in a practical elective course in February 2022 with the final year pharmacy students to evaluate the usefulness of various digital diabetes apps in the pharmaceutical care process, as previously reported [42]. The initial 25-item criteria were then discussed and re-evaluated by the authors (EO, BAS, SL, AD, and SS) and the list was refined to include the 16 most relevant criteria.

#### 2.1.2. General Characteristic Criteria

The criteria for the general characteristics of diabetes apps were extracted from the available literature [28,43,44,45], and by consulting a diabetes patient who was also an app developer (SK) to avoid bias. General characteristics ranged from very basic technical criteria (e.g., operating system, category, etc.) to more precise criteria, such as regulatory aspects (e.g., data protection, privacy policy, Conformitè Europëenne (CE) mark, etc.), financial options (e.g., cost, reimbursement), and the presence of scientific studies on selected apps.

#### 2.1.3. Patient Preference Criteria

Criteria for patient preferences related to active engagement with diabetes apps were also extracted from the available literature [28,43,44,45] and by consulting the diabetes patient/app developer (SK) and a pharmacist evaluator (EO) to ensure that the patient perspective was included. Patient preference criteria that were already covered by the pharmaceutical care criteria (e.g., reminder, food, activity functions) were excluded to avoid duplication.

### 2.2. Selecting the Diabetes Apps to Be Evaluated

All the authors selected ten widely used diabetes digital apps for evaluation based on a study by Kebede and Pischke, “Popular Diabetes Apps and the Impact of Diabetes App Use on Self-Care Behaviour” [46]; the DiGA directory [6]; and their availability and status in Germany [47].

### 2.3. Procedure for Evaluating the Apps Using the Criteria

The authors (EO, BAS, AD, SS) downloaded and installed all the apps on their smartphones (Android/iOS) and used the predefined evaluation criteria. Before evaluating the apps, the authors discussed the criteria (pharmaceutical-care-related criteria and general characteristics criteria) to ensure an understanding of all items listed for evaluation. All of the apps were assessed in their basic and premium versions, and their corresponding apps and app store websites were consulted to obtain information about certain functions. All the authors independently performed the app evaluation process.

The evaluation results were double-checked, compared, and discussed during six online meetings among evaluators. In total, there were five disagreements in the individual assessors’ evaluations of the apps, which are presented as follows:Pharmaceutical care criteria: drug information item. Some apps allow logging and tracking of medications but do not contain drug information, such as dosage, warnings, indications, and other aspects. As some of the criteria were not quite clear in regard to all evaluators, an additional literature review was performed in some cases so that a consensus could be reached.Pharmaceutical care criteria: drug selection item. The DiabetesConnect app allows the selection of a drug from a standard list of medications and not according to the latest guidelines. After double-checking, all assessors agreed that this was an important function and gave it a score of one for at least having the list of medications.Pharmaceutical care criteria: insulin bolus calculator item. The Dario Health bolus calculator was not found within the app. However, a discrepancy was noted as the app’s website stated that the app contained a bolus calculator. This was rechecked by the evaluators, who confirmed that this feature was not available in the app downloaded from the German app store. It was therefore given a score of zero.Pharmaceutical care criteria: communication item. Some apps (e.g., One Touch Reveal^®^) offered the possibility of exchanging information through SMS, but this was not considered within-app communication. Therefore, a score of zero was given to all apps with SMS capability.General characteristic: scientific studies on apps. Initially, no scientific studies were found for the Diabetes:M app. However, after double-checking the proceedings of a conference, a study on Diabetes:M was found. Therefore, it was given a ‘Yes’.

After a consensus was reached, the results were collated and summarized.

#### 2.3.1. Criteria Relevant to Pharmaceutical Care

The authors assessed the essential functions of apps related to the provision of pharmaceutical care by reusing anonymized data from real diabetes patients. The authors thoroughly assessed the defined criteria within the apps and searched on app store websites and on homepages of individual app developers in cases of discrepancies. The criteria were considered absent if they were not found in any of the above mentioned sources.

The criteria relevant to digital diabetes pharmaceutical care were scored based on an objective assessment as ‘Yes’ (1) and ‘No ‘(0) depending on the presence or absence of specific functions, respectively. The higher the total score, the more potentially helpful the app was in supporting pharmacists to provide pharmaceutical care to diabetes patients.

#### 2.3.2. General Characteristic Criteria

The authors (EO, BAS, AD, SK, and SS) also analyzed the apps for their general characteristics and made additional descriptive notes in cases of unique functionalities. To determine whether clinical studies supporting the use of the selected app had been published, EO, BAS, AD, and SS conducted additional PubMed and Google Scholar searches using the name of each selected app.

#### 2.3.3. Patient Preference Criteria

The diabetes patient/app developer (SK) evaluated the patient preference criteria for their presence or absence in the selected apps. These criteria were further re-checked by pharmacist evaluators (EO, BAS, AD, SS) to eliminate any discrepancies.

## 3. Results

### 3.1. App Evaluation Criteria

The 16 pharmaceutical-care-related criteria (Table 1) included: four criteria related to medication management (e.g., insulin dose and bolus calculation, drug information and selection, interaction checks); five related to adherence and non-pharmacological management (reminder/alert functions, warning functions, food and sport/activity functions, and personal notes), and seven related to interoperability and interaction/communication (e.g., communication and data exchange between patients and HCPs, interoperability with other devices/software, data storage and display, and the involvement of pharmacists). These functions, necessary to support the pharmaceutical care process, have been evaluated in all ten selected apps (Table 2). It is important to note that these criteria were neither exhaustive, nor were they used to evaluate the overall quality of the selected diabetes apps. Thirteen criteria were included in the list of general characteristics to be evaluated for diabetes apps (Table 3) and ten were included in the list of patient preferences for active engagement with diabetes apps (Table 4).

### 3.2. Selected Diabetes Apps

Of the ten diabetes apps selected for evaluation, six (mySugr, Diabetes:M, Contour™ Diabetes, OneTouch Reveal^®^, Dario Health, and DiabetesConnect) were selected based on a survey of the use of popular diabetes apps in Germany by Kebede and Pischke [46] (continuous glucose monitoring apps were excluded). The ESYSTA app was also included, as it was DiGA-approved for diabetes management at the time of evaluation [6]. The lumind app was selected based on its DiaDigital app quality certificate in Germany [47] and meala was chosen as a newer app developed by the same company. Glucose Buddy was selected as one of the most downloaded apps in February 2022. However, the authors of this paper are aware that there are also other potentially helpful and popular diabetes apps in addition to those chosen for this study.

### 3.3. Functions of the Apps Relevant to Pharmaceutical Care

The functions of diabetes digital apps supporting pharmacists in the pharmaceutical care process were evaluated for all ten selected apps (Table 2). Concerning the 16 criteria relevant to pharmaceutical care, Diabetes:M and mySugr met the most criteria, with each meeting a total of 11 criteria; Contour™ Diabetes, Dario Health, and OneTouch Reveal^®^ each met ten criteria; DiabetesConnect and ESYSTA both met nine criteria; Glucose Buddy and meala met eight and seven criteria, respectively, and lumind met only three criteria.

#### 3.3.1. Medication Management

The Diabetes:M, mySugr, and OneTouch Reveal^®^ apps include insulin bolus calculators to modify insulin doses according to individual needs (Table 2). The Insulin Mentor bolus calculator in the OneTouch Reveal^®^ app assists in calculating insulin bolus doses by considering multiple factors such as active insulin, blood glucose values, and carbohydrate intake. Similarly, the mySugr bolus calculator tool uses automatic algorithms and has been tested in clinical studies [48,49]. The Diabetes:M app bolus calculator provides extended bolus calculations by considering protein and fat intake in addition to carbohydrates. Overall, nearly all the evaluated apps lacked criteria relevant to medication management. None of the evaluated apps included drug information, drug interaction checking, or drug selection according to the latest guidelines (Table 2). Only DiabetesConnect allowed for drug selection from a list of standard medications rather than from the latest guidelines.

#### 3.3.2. Adherence and Non-Pharmacological Management

Overall, criteria that can support pharmacists in motivating patients for medication adherence and non-pharmacological management were met in most of the ten evaluated apps. All of the apps (except ESYSTA) included a reminder or alert function (Table 2) which reminded users about administering insulin, measuring blood glucose levels, doctor appointments, taking medications, or other essential tasks as set by the users. The reminders were in the form of text notifications or alarms or both. In addition, 80% of the apps could warn and alert users about hypo- or hyperglycemic events with either a color scheme (color codes indicating low, normal, and high blood glucose levels) or light (lumind). Most of the evaluated apps also included a food function (90%), a sports/activity function (80%), and/or the ability to add personal notes (90%).

#### 3.3.3. Interoperability and Interaction/Communication

Half of the ten evaluated apps, namely, ESYSTA, mySugr, Contour™ Diabetes, meala, and Dario Health possessed the ability to allow communication between patients and HCPs within the app (Table 2). The majority of apps (except lumind) allowed patient data to be exported to and shared with HCPs in various formats (PDF, CSV, etc.), and through various online platforms (e.g., email, WhatsApp). The OneTouch Reveal^®^ app allowed the exchange of information through the short message service (SMS) option.

Ninety percent of the evaluated apps (all but DiabetesConnect) provided the option to connect to other devices, such as blood glucose measuring devices, insulin pens, pumps, etc. (Table 2). However, the Dario Health and OneTouch Reveal^®^ apps only allowed connections with their related devices. Diabetes:M was the only app that included a smart watch compatibility. Most (80%) apps provided the option of synchronization with other apps or operating systems, and 90% could store and display all patient data graphically and statistically. Some apps (ESYSTA, Diabetes:M, DiabetesConnect, and Glucose Buddy) also allowed automatic data synchronization with an online web portal.

Although the functions promoting interoperability and exchange with HCPs were present in most apps, no app provided direct communication between patients and pharmacists (Table 2) or explicitly mentioned pharmacists as diabetes care providers.

### 3.4. General Characteristic of the Apps

The general characteristics of the evaluated diabetes apps are presented in Table 3. All the apps were developed for both Android and iOS platforms. Most (80%) of the apps fell into the medical category, with only lumind and meala falling into the health and fitness category. Important aspects of data protection and privacy policy were included in all the apps. Most of the apps (60%) contained the CE mark, which gives market authorization to the product throughout Europe.

Four of the evaluated apps (Contour™ Diabetes, lumind, meala, and OneTouch Reveal^®^) were cost-free. The remaining apps offered a trial version or free-of-cost access in the form of a basic version, with premium versions being associated with a variety of prices. As of March 2022, only the ESYSTA and mySugr apps could be reimbursed by the statutory health insurance in the German healthcare system. Advertising was not present in premium (paid) versions of any of the apps.

Studies providing evidence on app effectiveness and other patient-reported outcomes were found for mySugr [50,51], Glucose Buddy [51,52], Diabetes:M [53], OneTouch Reveal^®^ [54], ESYSTA [55], Dario Health [56], and Contour™ Diabetes [57].

All the apps, except meala, could be accessed offline. Registration or login was required for accessing most of the apps; Diabetes:M could be used without logging in but with limited functionality, whereas lumind and meala could be used with their full functionality without the need to log in. Only Glucose Buddy could be logged into with an existing account (e.g., Facebook, Google, etc.).

### 3.5. Patients’ Preferences and Other Special Functions

The availability of patients’ preferences for active engagement with the ten selected diabetes apps were also evaluated (Table 4). These criteria included peer support (20% of evaluated apps), swarm knowledge (10%), training and educational material (50%), analysis of blood glucose values and therapy recommendations (80%), usability and appealing design (70%), app accessibility in the case of visual and hearing impairments (10%), the ability to set individual target ranges (70%) or carbohydrate units (50%), multiple-profile management (10%), and the ability to share data with followers (10%).

The evaluated apps also included some unique functions of interest, including motivation through gamification and challenges (mySugr), emergency contact functions (Contour™ Diabetes and Dario Health), estimated HbA1C function (ESYSTA, mySugr, and Dario Health), food databases (Diabetes:M and Glucose Buddy), personal diabetes coaching (Dario Health, mySugr, and Glucose Buddy), and a smart assistant function (Diabetes:M). Blood pressure, weight management, and other laboratory data could also be stored in some apps in addition to diabetes-related data (Dario Health, Diabetes:M, and DiabetesConnect), hence offering an option for the management of multiple chronic diseases.

The two apps belonging to the health and fitness category (i.e., meala and lumind) offered additional unique functions. In particular, the meala app had functions to recognize meals and avoid mistakes when estimating carbohydrates, and the lumind Habitat app retrieved data on blood glucose levels from compatible meters and converted them into sounds, light, and colors, which is especially useful for patients with hearing aids and visual impairment.

## 4. Discussion

Pharmacists have successfully initiated various pharmaceutical care programs for patients with diabetes by implementing pharmacological and non-pharmacological interventions [29,58]. Their success has become even more critical in the post-pandemic era in relation to lowering the risks of diabetes-related acute complications and subsequent hospitalizations [59]. As the use of diabetes apps may help pharmacists improve pharmaceutical care and outcomes in diabetes patients, it is important that pharmacists are aware of the functions of such apps when recommending their use.

The aim of this study was to provide information about the functions of diabetes apps related to providing pharmaceutical care services to diabetes patients. To accomplish this, in this study we evaluated the functions of selected diabetes apps based on criteria relevant to pharmaceutical care, as well as criteria relevant to technological needs and the preferences of patients. We also addressed the shortcomings of the apps to unfold the full potential for patients’ interaction with HCPs, such as direct communication with HCPs (i.e., pharmacists) and the lack of medication management functions, which is particularly important for polypharmacy in patients with type 2 diabetes.

The first major question to be answered in this study was ‘What functions of diabetes apps support the provision of pharmaceutical care?’ In order to answer this, the criteria to evaluate such functions first needed to be developed. Based on the literature, a list of essential criteria based on the important pharmaceutical care interventions for diabetes management was compiled. These criteria were then tested and refined to include the 16 most relevant criteria from a pharmaceutical care perspective. Criteria for evaluating the general characteristics and patient preferences for diabetes apps were also developed based on the available literature and consultation with appropriate individuals. These novel evaluation criteria were then used to evaluate the selected diabetes apps, and could be used in future evaluations of diabetes apps and refined to evaluate the functions of digital apps for other health conditions.

The second major question to be answered was ‘Which currently available diabetes apps fulfil the criteria relevant to pharmaceutical care?’ The results of this study revealed that most of the evaluated digital apps for diabetes patients could help provide pharmaceutical care, as eight of the ten apps integrated >50% of the criteria related to pharmaceutical care. The total scores ranged from 3 to 11 out of a maximum score of 16; 80% of the apps had a score of ≥8, and only 20% of apps had a score of <8. The pharmaceutical care functions that were lacking in these apps may be due to the current lack of involvement of pharmacists (unlike other members of the multidisciplinary diabetes care team) in the initial design of diabetes apps.

The pharmaceutical care criteria comprised three major categories, the first of which was ‘medication management’. Medication logging and tracking functions were available in some of the apps. However, medication management functions, particularly for patients with type 2 diabetes, such as drug information, drug selection according to guidelines, and checking for drug interactions, were not provided by any app. A 2019 study also reported a general lack of medication management functions in diabetes apps [60]. Four of the ten evaluated diabetes apps were integrated with a bolus insulin calculator, simplifying the complex task of insulin dose calculation, a very useful function for patients prescribed with insulin. However, they need to be carefully and cautiously employed, as mistakes resulting from user errors and/or software errors can lead to serious consequences [61]. Moreover, the lack of important functions to validate insulin dose calculations could result in harmful dose recommendations [26] and adverse events [17]. Pharmacists can have an important role in minimizing these potential errors by carefully considering all the user- and app-related factors. ‘Adherence/non-pharmacological management’ was the second major category of pharmaceutical care criteria. These were the most prevalent functions of the selected diabetes apps. Of the ten evaluated digital apps, 60% met all five criteria in this category. Ninety percent of the apps included a reminder/alert function, personal notes, and/or food function. In addition, 80% of the apps had a warning function and a physical activity logging function. Evidence suggests that app-based adherence interventions for patients with diabetes have resulted in decreasing HbA1C levels by improving adherence behaviors to medications, diet, and exercise [62].

The final major category of pharmaceutical care criteria was ‘interoperability/communication’. Communication between patients and HCPs through mHealth apps serves as an alternative to in-person clinical visits and face-to-face contact. Diabetes care can benefit greatly from patient–provider contact facilitated by apps and web portals [63]. Clinical outcomes and medication adherence among diabetes patients have been reported to be improved by pharmacist-led follow-up interventions that involved the simplest methods of telecommunication, such as telephone calls and text messages [30,31,32]. Although half of the evaluated apps (ESYSTA, mySugr, Contour™ Diabetes, meala, and Dario Health) allowed for communication with HCPs inside the app to ensure real-time support and feedback for diabetic patients, none of the apps allowed for communication specifically with pharmacists. Within-app communication is also important as regular follow-up of patients by HCPs has been shown to help prevent long-term complications of diabetes [64]. High-frequency HCP feedback through diabetes apps resulted in a mean HbA1C reduction of 1.12% compared with less frequent feedback (0.33%) and no feedback (0.24%) in a systematic review and meta-analysis of 21 randomized trials that evaluated the effect of diabetes apps [10]. Similar findings have been reported by other studies focusing on the effect of real-time HCP support and communication through mobile app interventions in diabetes management [63,65].

The third and final question to be answered in this study was ‘‘What additional app functions should pharmacists consider to satisfy the technological and health-related needs, and preferences of diabetes patients?’ The general technological characteristics of the evaluated diabetes apps were assessed descriptively to ensure an understanding of their usability, regulatory requisites, cost, reimbursement, and outcomes as published in studies in the scientific literature. It is also important for pharmacists to review these aspects before recommending apps to their patients. Most of the apps belonged to the medical category and contained a CE mark, which means that they met the standards set by the Medical Device Regulation policy standards [27]. However, the presence of a CE mark does not mean that the app has been tested for accuracy and clinical outcomes [66]. Disclosures of privacy policies and data protection policies were present in all apps. Only four of the apps were free to use; however, nearly all other apps offered a test or trial version. Reimbursement through statutory health insurance was only possible for the ESYSTA and mySugr apps at the time of evaluation.

Studies in the scientific literature were found for 70% of the apps, with a range of beneficial study outcomes. Two broader categories of study outcomes were clinical outcomes (e.g., impact on HbA1C, glycemic variability, etc.) and self-reported outcomes (e.g., self-care, quality of life, patient empowerment, satisfaction, engagement, etc.). A study by Drincic et al. reported similar findings regarding safety and efficacy outcomes for diabetes apps based on the published literature [67].

Patients’ preferences for certain functions are very important in order to foster their long-term and consistent engagement with diabetes apps [44] and, in turn, guide HCPs to recommend suitable apps [28]. We believed it was important to evaluate diabetes apps according to patients’ perspectives, since the insufficient assessment of end-users’ expectations has been related to a low level of app adoption and use [44]. In addition to the most used and preferred functions of blood glucose tracking, carbohydrate/calorie counting, and physical activity tracking [45,68], the present study also included more sophisticated functions. Training and educational material, analysis of blood glucose values and therapy recommendations, usability and appealing design, individual target range settings, and individual carbohydrate unit functions were available in most of the assessed apps. However, the ability to share data with followers, access to peer support and swarm knowledge, multiple profile management, and app accessibility in the case of visual and hearing impairments were found in very few diabetes apps.

In the context of “apps on prescription” (DiGA), pharmacists in Germany are often confronted by patients with questions about mHealth apps and their effectiveness. Therefore, this review of app functionality could help practicing pharmacists to become familiar with the essential aspects of apps and to become aware of the need to educate and counsel their patients about practical app usage during patient encounters. Although the focus of this study was diabetes apps, the findings from this study can be applied to evaluate the pharmaceutical-care-related functions of other disease-specific apps.

The study has several limitations. The general characteristics of the diabetes apps were evaluated only on a descriptive level, as a more detailed assessment was outside the scope of this study. Once each assessor had independently evaluated the various app criteria, the assessors met to double-check, compare, and discuss their results and resolve any discrepancies in their evaluations via a consensus. The inter-rater variability among all four authors was not determined and therefore the lack of statistical analyses of the inter-rater reliability of the assessors is a potential limitation of the study. In addition, some of the personal preference criteria were vague and subjective, i.e., the criteria ‘Is the app easy to understand and is the UI/UX design appealing?’, and the results may have been influenced by the experience and knowledge of the diabetes patient who was also an app developer and assessor. It may have been more appropriate to have this criterion evaluated by other diabetes patients rather than by individuals who were familiar with such apps. Furthermore, this review is limited by the selection of only ten diabetes apps for evaluation, which could limit the generalizability of our findings. Moreover, only one diabetes patient was involved in the evaluation process; however, this individual had profound knowledge and experience in designing and developing apps. Finally, app functions are continuously updated, and there is a possibility that many of the apps will have been upgraded with new functions by the time the results of this study are available.

## 5. Conclusions

Our evaluation showed that diabetes apps were equipped with the necessary functions to support pharmacists and other HCPs in providing pharmaceutical care services to patients with diabetes. Nonetheless, improvement in their functions is needed as they often lacked medication management functions. Furthermore, the careful supervision of diabetes self-management through apps is necessary in order to amplify app effectiveness, increase app adherence, and mitigate the risks associated with improper use. On the other hand, the provision of app-integrated pharmaceutical care could provide an unprecedented opportunity for pharmacists to develop active interactions with their patients through remote monitoring and intervention, which is currently not possible. The evaluation of digital diabetes apps shows that apps can be powerful tools for pharmaceutical care; however, they are still broadly underutilized by pharmacists. Hence, diabetes app providers should recognize pharmacists’ expertise and specifically include them alongside other clinicians in customized versions of diabetes apps. In addition, a direct exchange through the app between patients and pharmacists (i.e., a chat function) would further enhance the pharmaceutical care process and help improve diabetes outcomes.

## Figures and Tables

**Table 1 ijerph-20-00064-t001:** Description of criteria relevant to digital pharmaceutical care for patients with diabetes.

Criteria	Description of the Ability of the App
**Medication management**	
1. Drug information	Able to provide information about drugs, such as indication, dosage, warnings, and other aspects
2. Drug selection	Able to help select drugs according to the latest guidelines
3. Insulin bolus calculation	Able to calculate insulin bolus doses
4. Interaction check (type 2)	Able to check drug interactions
**Adherence/Non-pharmacological management**	
5. Reminder/alert	Able to remind or alert users on insulin administration, blood glucose measurements, doctor appointments, etc.
6. Warning function	Able to notify or warn users about hypo-or hyperglycemic events in real time
7. Food function	Able to enter additional different foods manually or via bar code scanning, selecting from databases, taking pictures, etc.
8. Sports/activity function	Able to log sports or other physical activities
9. Personal notes	Able to add personal notes when desired
**Interoperability and interaction/communication**	
10. Communication (between patient and HCPs)	Able to communicate with HCPs (within the app)
11. Possible to exchange data with HCPs	Able to retrieve and share data with healthcare professionals
12. Possible to connect to devices	Able to connect to other devices, such as blood glucose measuring devices, insulin pens, pumps, etc.
13. Smart watch compatibility	Compatible with smart watches and smartwatch apps
14. Synchronization option	Able to synchronize between different apps and operating systems
15. Data storage and display	Able to store and display graphical and statistical data
16. Pharmacist involvement	Able to allow pharmacists to intervene with a pharmacist-specific dashboard

HCP, healthcare professional.

**Table 2 ijerph-20-00064-t002:** Evaluation scores for mobile health apps for diabetes patients based on criteria relevant to digital pharmaceutical care for diabetes patients.

Criteria Relevant to Digital Diabetes Pharmaceutical Care ^a^	Diabetes Mobile Health Apps									
	ESYSTA	mySugr	Diabtes: M	Contour^TM^ Diabetes	Dario Health	Diabetes Connect	Glucose Buddy	lumind	meala	OneTouch Reveal^®^
**Medication management**										
1. Drug information	0	0	0	0	0	0	0	0	0	0
2. Drug selection	0	0	0	0	0	1 ^b^	0	0	0	0
3. Insulin bolus calculation	0	1	1	0	0	0	0	0	0	1
4. Interaction check (type 2)	0	0	0	0	0	0	0	0	0	0
**Adherence/non-pharmacological management**										
5. Reminder/alert	0	1	1	1	1	1	1	1	1	1
6. Warning function	1	1	1	1	1	1	0	1	0	1
7. Food function	1	1	1	1	1	1	1	0	1	1
8.Sports/activity function	1	1	1	1	1	1	1	0	0	1
9. Personal notes	1	1	1	1	1	1	1	0	1	1
**Interoperability and interaction/communication**										
10. Communication (between patients and HCPs)	1	1	0	1	1	0	0	0	1	0
11. Possible to exchange data with HCPs	1	1	1	1	1	1	1	0	1	1
12. Possible to connect to devices	1	1	1	1	1 ^c^	0	1	1	1	1 ^c^
13. Smart watch compatibility	0	0	1	0	0	0	0	0	0	0
14. Synchronization option	1	1	1	1	1	1	1	0	0	1
15. Data storage and display	1	1	1	1	1	1	1	0	1	1
16. Pharmacist involvement	0	0	0	0	0	0	0	0	0	0
Total score (maximum 16)	**9**	**11**	**11**	**10**	**10**	**9**	**8**	**3**	**7**	**10**

HCP, healthcare professional; 1 indicates that the individual feature/function was found within the app, on app store websites, or on the homepages of individual app developers; 0 indicates that the individual feature/function was not found. ^a^ See Table 1 for a description of the criteria. ^b^ Drug selection according to a standard list instead of the latest guidelines. ^c^ Only their related devices.

**Table 3 ijerph-20-00064-t003:** General characteristic criteria of mobile health apps for diabetes patients.

General Characteristics	Diabetes Mobile Health Apps									
	ESYSTA	mySugr	Diabetes: M	Contour^TM^ Diabetes	Dario Health	Diabetes Connect	Glucose Buddy	lumind	meala	OneTouch Reveal^®^
Category	Medical	Medical	Medical	Medical	Medical	Medical	Medical	Health & fitness	Health & fitness	Medical
Android/iOS operating system	Yes	Yes	Yes	Yes	Yes	Yes	Yes	Yes	Yes	Yes
Data protection	Yes	Yes	Yes	Yes	Yes	Yes	Yes	Yes	Yes	Yes
Privacy policy	Yes	Yes	Yes	Yes	Yes	Yes	Yes	Yes	Yes	Yes
Medical device classification	Yes	Yes	Yes	Yes	No	No	Yes	No	No	Yes
Cost	Yes	Yes ^a^	Yes	Free	Yes	Yes	Yes	Free	Free	Free
Trial version/test version	Yes	Yes	Yes	NA *	Yes	Yes	Yes	NA *	NA *	NA *
Reimbursement	Yes	Yes	No	NA *	No	No	No	NA *	NA *	NA *
Advertising	No	No	No ^b^	No	No	No	No	No	No	No
Studies conducted with apps	Yes	Yes	Yes	Yes	Yes	No	Yes	No	No	Yes
Offline availability	Yes	Yes	Yes	Yes	Yes	Yes	Yes	Yes	No	Yes
Usable without login/registration	No	No	Yes ^c^	No	No	No	No	Yes	Yes	No
Login possible with an existing account	No	No	No	No	No	No	Yes	No	No	No

* NA, not applicable, as app was cost-free. ^a^ Free when used with Accu-Chek^®^ devices. ^b^ Only the basic version included advertisements. ^c^ Could be used without logging in but with limited functionality.

**Table 4 ijerph-20-00064-t004:** Patients’ preferences for active engagement with diabetes apps.

Patient Preferences (Description)	Diabetes Mobile Health App									
	ESYSTA	mySugr	Diabetes:M	Contour™ Diabetes	Dario Health	Diabetes-Connect	Glucose Buddy	lumind	meala	One Touch Reveal^®^
**Peer support/exchange with other patients**										
Is there a way to exchange ideas with other patients (e.g., forums)?			✓						✓	
**Swarm knowledge/insight into others’ experiences**										
Is it possible to view the experiences of other patients and learn from them (e.g., CGM values)?									✓	
**Training and information materials**										
Are information/training materials available to improve one’s knowledge and, if necessary, healthcare?		✓	✓		✓		✓		✓	
**Analysis of blood glucose values**										
Are the blood glucose values entered automatically analyzed and are therapy recommendations given for optimization?	✓	✓	✓	✓	✓	✓	✓			✓
**UI/UX design**										
Is the app easy to understand and is the UI/UX design appealing?		✓		✓	✓		✓	✓	✓	✓
**Accessibility**										
Is the app accessible to people with visual or hearing impairment?								✓		
**Individual target area**										
Can the target range be set manually?	✓	✓	✓	✓	✓	✓	✓			
**Individual carbohydrate units (e.g., bread units, carbohydrate units/g)**										
Can the unit of carbohydrates be individually adjusted?	✓	✓	✓	✓		✓				
**Management of multiple profiles with one account**										
Can one account manage multiple blood glucose profiles?			✓							
**Follower function**										
Can data be shared with family and friends as “followers”?								✓		

CGM continuous glucose monitoring; UI user interface; UX user experience.

## Data Availability

Not applicable.

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
