# Peer review of "Functions of mHealth Diabetes Apps That Enable the Provision of Pharmaceutical Care: Criteria Development and Evaluation of Popular Apps"

_ijerph, 2022, doi:10.3390/ijerph20010064_

Round 1

Reviewer 1 Report

The authors present a heuristic evaluation of several medication adherence mobile apps for diabetes patients. The study has been well conducted and presented. I have the following comments that can be used to improve the paper.

-The authors have not explicitly presented a solid contribution of their work. It is not clear what is the importance of answering the three questions that have been presented in the introduction section. The authors can improve the discussion section by explicitly focusing on this aspect. 

- Table 4 has some very vague / subjective criteria. For example, UI/UX of the app was measured through ease of use and attractiveness of the app. These are very subjective and I think that an app developer is perhaps not the best choice for this purpose. Authors can discuss this limitation in the paper. 

- The title of the paper does not correspond with the content of the paper, which is mainly a heuristic evaluation study. 

Author Response

Dear reviewer,

Please find attached our word document Answer to the Reviewer 1 Comments dated 17.12. 2022. (please disregard the version of 15.12.22)

Thank you very much for your valuable comments. We hope that we have addressed all your comments which helped us revise the article.

With the comments now inserted and with the revised text, we hope to have improved the quality of our article so that it can be considered for publication.

Thank you.

Kind regards,

Dr. Emina Obarcanin

Reviewer 2 Report

Thank you for the opportunity to review this interesting paper “Features of mHealth diabetes apps that enable the provision of pharmaceutical care and improved patient outcomes: criteria development and evaluation of popular apps”. Authors developed three criteria to evaluate the quality of diabetes apps currently available in the market, chose the 10 most popular apps for quality evaluation, and determined the lack of medication management features among most popular apps. Therefore, authors recommended the involvement of pharmacists during the app design and emphasized the importance of patient healthcare provider communication. Overall, this was a well-written paper and added its value to the current literature pool on the quality evaluation and improvement of mHealth apps. However, I have several suggestions.

Major:

1.       This study did not have direct evidence linking mHealth apps to improved patient outcomes. Therefore, the title: “Features of mHealth diabetes apps that enable the provision of pharmaceutical care and improved patient outcomes: criteria development and evaluation of popular apps” might not be accurate. I would suggest revising the title. For example: Features of mHealth diabetes apps that enable the provision of pharmaceutical care: criteria development and popular apps evaluation.

2.       This study mainly addressed three criteria development of mHealth apps based on different features: criteria to pharmaceutical care, general characteristic criteria, and patient preference criteria. However, in the introduction section, authors addressed the background of these major features and criteria development very superficially [I can see only one paragraph in the introduction section addressing such features (the fifth paragraph, Line 76-84)]. In addition, even within this paragraph, the features are not addressed clearly. For example, “current diabetes apps, have a variety of features… However, they lack some clinically relevant features… many reviews have evaluated the functional features” I read with some confusion. A variety of features, are these features consider function features? Do clinically relevant features belong to function features. How to categorize these features? What is the difference between function features vs. clinical relevant features vs. features in general? Please address clearly. In addition, since criteria development and app quality evaluation are the focus of this manuscript, such criteria and mHealth feature required to be addressed intensely in the introduction section.

3.       In method, Line 132, procedure for evaluating the apps using the criteria: “The evaluation results were double-checked, compared, and discussed during six online meetings among reviewers, and discrepancies were resolved. Finally, the results were collated and summarized after reaching a consensus.” Since four authors evaluated separately, I believe simply addressing "discrepancies were resolved… reaching a consensus” might not enough. It is better to address the inter-rater variability (for example using kappa statistics to determine the inter-rater variability among all 4 authors)

Minor

1.       Line 25, abbreviation HCPs was shown the first time, should be interpreted.

2.       Line 37, “the” should be “The”

3.       Line 118: “such as e regulatory aspects” should be “such as regulatory aspects”

4.       Line 204: “…several clinical studies” need references

5.       Line 233: “All apps (except lumind)…” suggest revising as “The majority of apps…”

6.       Line 244: “…with an online web portal” please add the period. “…with an online web portal.”

Author Response

Dear reviewer,

thank you very much for the reviewing and excellent comments which helped us improve our article. Please see our answers to your comments attached.

Kind regards,

Dr. Emina Obarcanin
